# Automated Liquid–Liquid Displacement Porometry (LLDP) for the Non-Destructive Characterization of Ultrapure Water Purification Filtration Devices

**DOI:** 10.3390/membranes13070660

**Published:** 2023-07-11

**Authors:** René I. Peinador, Daniel Darbouret, Christophe Paragot, José I. Calvo

**Affiliations:** 1Institut de la Filtration et des Techniques Séparatives (IFTS), Rue Marcel Pagnol, 47510 Foulayronnes, France; 2Millipore S.A.S., (affiliate of Merck KGaA, Darmstadt, Germany), 78280 Guyancourt, France; daniel.darbouret@merckgroup.com (D.D.); christophe.paragot@merckgroup.com (C.P.); 3Departamento de Física Aplicada, ETSIIAA, Universidad de Valladolid, 34071 Palencia, Spain; 4Institute of Sustainable Processes (ISP), Dr. Mergelina s/n, 47071 Valladolid, Spain

**Keywords:** membrane characterization, pore size distribution (PSD), filtration devices, liquid–liquid displacement porometry (LLDP), submicronic efficiency

## Abstract

This scientific publication presents a novel modification of the liquid–liquid displacement porosimetry (LLDP) method, aiming for the non-destructive automated analysis of water purification membrane filtration devices in the microfiltration (MF) and ultrafiltration (UF) range. The technical adaptation of LLDP enables the direct in-line porosimetric analysis of commercial filtration devices, avoiding the filtration devices’ destruction. Six commercially available filtration devices with polyethersulfone (PES) and polysulfone (PS) membranes were studied using an improved device developed by the IFTS, which was based on a commercial LLDP instrument. The filtration devices were evaluated in three different configurations: flat disks, hollow fibers, and pleated membranes. The results obtained using the proposed method were compared with other characterization techniques, including submicronic efficiency retention, image analysis of scanning electron microscopy (SEM), and gas–liquid displacement porosimetry (GLDP). The comparison of the results demonstrated that the proposed method accurately determined the porosimetric characteristics of the filters. It proved to be a precise technique for the non-destructive in-line evaluation of filter performance, as well as for periodic quality control and the fouling degree assessment of commercial filtration devices. This modified LLDP approach offers significant potential for the advanced characterization and quality assessment of water purification membrane filtration devices, contributing to improved understanding and optimization of their performance.

## 1. Introduction

Membrane processes, particularly water sterilization, have become crucial in various applications, ranging from laboratory use to industrial processes. Initially, filtration for sterilization purposes was carried out in small quantities, primarily for laboratory applications and injectable medicines, using dead-end mode filtration [1]. However, as these processes became integrated into industrial processes, the need for larger membrane surface areas to treat larger volumes of fluid more efficiently arose. Additionally, the objective of minimizing membrane fouling led to the adoption of the frequent crossflow operation.

The early membrane filters, initially referred to as ultrafilters [2], exhibited porosity characteristics within the range now known as microfiltration (MF). Some filters with pore sizes falling within the category of ultrafiltration (UF) were also developed. Despite the extensive development of synthetic membranes and the outstanding increase in membrane applications over the past century, water sterilization remains one of the most prevalent uses of MF membranes. Consequently, the use of 0.2–0.22 µm MF filters prior to bottling is recommended to prevent bacterial contamination of water and various industrial liquids, such as pharmaceutical formulations, fruit juices, and beverages. In the case of MF filters used for water and liquid filtration, their performance must be evaluated based on the retention of *Brevundimonas diminuta*, as per the ASTM F838-20 standard [3].

Initially, membranes were produced as flat pieces, limiting the filtration area. However, fabrication methods like phase inversion were well suited for obtaining flat membrane sheets. To increase the filtration area, the next step involved incorporating as many flat sheets as possible into the filtration device or module, leading to the development of various packing configurations, such as spiral wound modules. Spiral winding involves winding a set of flat membranes separated by fabric layers that act as transporters and generators of turbulence for the feed and permeate solutions. These flat membranes are wound around a central perforated tube, resulting in a significantly larger membrane surface area while reducing energy costs. However, this configuration also increases the fouling of the module and makes cleaning more challenging.

The introduction of the hollow fiber configuration was a significant advancement, utilizing the spinning system proposed by Mahon at Dow Chemical in the 1960s [4,5]. Hollow fiber membranes offer an increased filtration area per unit volume, provide mechanical self-support, and are easy to handle [6,7]. Consequently, the use of hollow fibers has grown tremendously over the past 6–7 decades [8,9,10,11,12,13,14,15,16].

The performance of hollow fibers, in terms of permeability and selectivity, depends on various structural aspects determined by the manufacturing process. The key factors influencing their potential applications include the physicochemical and mechanical properties of the material used, the thickness of the active separation layer determining selectivity and permeability, and the size and distribution of the pores within the fiber.

To determine the membrane structure and pore characteristics, various characterization methods, known as porometries or porosimetries, are employed. These include gas adsorption/desorption (GAD), mercury porosimetry (HgP), thermoporometry (ThP), permporometry (PmP), gas–liquid displacement porosimetry/liquid–liquid displacement porosimetry (GLDP/LLDP), and evapoporometry (EP) [17]. These techniques provide indirect information on the membrane surface and pore sizes. Some methods offer information on all the pores in the sample, while others focus on the active layer of the membrane. It is important to select a porosimetric technique suitable for analyzing the active layer’s pores, as they ultimately determine the membrane’s selectivity.

However, most porosimetric techniques are designed for analyzing small samples, such as flat discs or cut portions introduced into the measuring cell. These methods are not suitable for analyzing a complete membrane filtration device without destroying it. Analyzing hollow fiber membranes would involve joining a sufficiently large set of fibers and sealing them together in a module suitable for porosimetric analysis. This approach provides average information on the set of fibers.

Ideally, to characterize a membrane filtration device, such as those used for ultrapure water, a device (porometer) should be capable of non-destructively obtaining information about the pores in the filtration device. Bubble point-derived methods, particularly GLDP, which measure the fluid flow through the membrane, can be used to characterize commercial modules by replacing the usual measuring cell with the filtration devices being analyzed. However, the high air flow rates provided by these filtration devices often exceed the measurement capabilities of commercial porometers, which are designed for smaller flow rates and sample areas. Consequently, the typical approach is to destroy the filtration device’s container to extract several pieces of flat membrane for analysis, raising concerns about potential sample falsification during the extraction process.

In a previous study [18], a liquid displacement porometer (FFP; model PRM-8710, from IFTS, Foulayronnes, France) was adapted to analyze commercial filtration devices non-destructively. The porometer was modified to work with commercial filtration devices, allowing the flow data to be obtained by applying pressure using the porometer and measuring the resulting flow using a balance. This information was used to obtain the pore size distribution (PSD) of the filtration device. However, this modification involved manual acquisition of flow data, which decreased the accuracy and reliability. Nonetheless, the adapted porometer was proved to be suitable for analyzing several commercial filtration devices.

In the present work, an improved version of the PRM-8710 porometer, which is capable of automated and operator-controlled analysis of commercial filtration devices, was used for non-destructive characterization purposes. The device operates in LLDP mode, since GLDP operation requires measuring very high air fluxes. Various commercial MF/UF filtration devices designed for water purification and bacterial removal were analyzed using this porometer. The results were compared with nominal values and with results obtained from conventional GLDP and SEM image analyses for some filtration devices. It is worth mentioning that the filtration devices’ destruction and membrane extraction were necessary for the SEM imaging and obtaining comparative data via GLDP.

The primary advantage of the porosimeter developed in this study is its non-destructive nature, eliminating the need to destroy filtration devices when analyzing the membranes’ structural properties. Additionally, the porosimeter incorporates automatic control, data acquisition, and data processing functionalities, enhancing its usability and efficiency. The modified version of the IFTS commercial porometer successfully characterized various membrane types, including hollow fiber, pleated, and flat disk-based filtration devices. This demonstrates the versatility and high performance of the newly proposed porometer, expanding its applicability in membrane characterization.

## 2. LLDP/GLDP Principles 

Fluid–fluid displacement porometry (FFDP) is a porosimetric technique based on the original bubble point method proposed by Bechhold in 1908–1909 [19] and adapted under the ASTM F316-03(2011) standard [20]. The technique has been very often known in the scientific literature as capillary flow porometry (CFP), a name which points to the fact that the analysis is based on flow determination through a membrane filter and assuming capillary pores inside it.

The technique, which is based on the use of the Young–Laplace equation for modeling the interface between two immiscible fluids inside a pore, is based on completely wetting the sample to be analyzed with a suitable liquid (which guarantees a zero-contact angle between the wetting liquid and the inside of the membrane pores). Subsequently, the sample is subjected to pressure applied by means of another fluid, which is immiscible with the wetting liquid. Depending on the state of aggregation of the second fluid (pushing fluid), we will talk about GLDP or LLDP, depending on whether the pushing fluid is a gas or another liquid, respectively.

In the present work, both the LLDP and GLDP techniques were used. Firstly, demonstrating that the usual LLDP equipment can be adapted for the non-destructive characterization of membrane filtration devices, which is the main objective of this study. However, to be sure of the accuracy of the results obtained from such non-destructive LLDP characterization, these results have been compared with the values obtained via GLDP after extracting the membranes from the filtration device. In both cases, the displacing (or pushing) fluid (immiscible with the wetting liquid) is pressurized into the cell containing the membrane previously soaked in the wetting liquid. As the applied pressure is gradually increased, the smaller pores are emptied of the wetting liquid and replaced with the displacing fluid which begins to permeate the membrane (see Figure 1).

The measurement of the flow through the membrane of this pushing fluid, as a function of the applied transmembrane pressure, allows us to determine the number and size of the pores successively opened at each stage of the pressure increase.

Thus, for each applied pressure, the size of the pores that become opened (those which are emptied of the wetting fluid and replaced with the pushing fluid) is given by the well-known Young–Laplace equation, which, for cylindrical capillaries, is written as:(1)$$\Delta p=\frac{4\gamma cos\theta }{d_{p}}$$
where *γ (N/m)* is the surface tension of the wetting liquid and *θ* is the contact angle between the liquid and capillary wall.

These methods require maintaining the contact angle between both fluids and the solid interface as low as possible to obtain reliable results. Ideally, the contact angle should be nil to ensure complete wetting. In practice, what is done is to assume a complete wetting of the sample due to a proper selection of wetting liquid; therefore, we can assimilate a value of *cos θ =* 1 in Equation (1). Certainly, for this purpose, it is essential to choose a convenient wetting liquid that provides the assumed perfect wetting.

On the other hand, as we increase the pressure and, consequently, we empty the smaller pores of the membrane of the wetting liquid, the permeability of the membrane in relation to the permeating liquid (what we called above the pushing liquid) becomes greater, so that we can evaluate the successive increases in the permeability at each stage of the pressure increase. These increases can be converted into dimensionless percentage increments with respect to the total or final permeability, *L_tot_*, using the following expression:(2)$$\Delta L_{k}=\frac{L_{k}-L_{k-1}\;}{L_{tot}}$$
where *L_k_* is the permeability of the *k*-th experimental point (*k =* 1, 2 … *i*) and *L_tot_* is the final (or asymptotic) permeability, corresponding to the last measured point (i.e., *k = i*). Representing these values of the percentile permeability increase versus the pore radius opened at each pressure stage, we obtain a pore size distribution that we can consider in terms of permeability (permeability PSD).

If we want further to convert this PSD into information about the number of pores opened in each of these stages, we must assume a transport model inside the pores. In the usual working conditions, the most suitable transport model is the Hagen–Poiseuille model for convective flow inside cylindrical capillary pores, so that we can assign a number of pores to each of the permeability increment values given by the following expression [21]:(3)$$\;n_{k}\left(d\right)=\frac{256\eta l}{\pi d_{k}^{5}}\Delta L_{k}$$
where η is the dynamic viscosity of the displacing liquid; l is the length of the pore; and $d_{k}$ is the pore diameter of the pores opened up to the *k*-th step. Finally, it must be noted that l accounts for the pore length value, which is equal to the active layer thickness for the usual case of asymmetric membranes, while l matches the whole membrane thickness in the case of symmetric ones. 

While, in the case of GLDP, the decision about which transport model has to be used is somehow more complicated, the Knudsen (molecular) flow can be considered along with the convective Hagen–Poiseuille model, depending on the relation between the pore size and mean molecular free pathway of the gas molecules moving inside the pores. In such cases, transitional modeling should be used to account for such situations [22], with more reliability in the results. Even so, in many commercial devices, the Hagen–Poiseuille model for convective flow is assumed in all the calculations.

## 3. Experimental

### 3.1. Membranes

This study describes the analysis of six commercially available membrane filtration devices designed for water purification applications, specifically targeted to the production of ultrapure water. These filtration devices are widely utilized in various laboratory applications, such as cell culture and microbiology. The filtration devices’ configurations include stacked flat discs, pleated flat membranes, and hollow fiber membranes, as outlined in Table 1. The membranes within these filtration devices are fabricated from nylon, polyethersulfone (PES), and polysulfone (PS), providing a high degree of chemical compatibility with a wide range of chemical solutions. Notably, all the membranes exhibit a unique structure characterized by conical-shaped pores. This structural design facilitates efficient water filtration at a higher flow rate while maintaining low differential pressure across the membrane.

Five of the analyzed filtration devices fall within the microfiltration (MF) range, making them suitable for removing particulate matter and microorganisms from water. Additionally, one filtration devices belonging to the ultrafiltration (UF) range was included in the study to assess the applicability of the characterization setup to membranes with smaller pore sizes. All the experimental investigations for characterizing the filtration devices filters were conducted at the IFTS laboratory in France.

The nominal information provided by the manufacturers regarding the membranes used in the filtration devices is outlined in Table 1. Specific details, such as the commercial names and traders, are not disclosed in this study.

For the LLDP analysis (non-destructive method), the filtration devices were placed connected to specifically designed adapters installed in the IFTS porometer used. In all the non-destructive characterizations performed (submicronic efficiency, LLDP and water permeability test), the filtration devices were operated in a dead-end filtration, and in the case of those containing fibers (A, E and F), they operated in an out–in configuration.

Some of the filtration devices (one of each membrane configuration) were dismantled and the membranes extracted to be tested with other characterization methods for the sake of comparison and validation of the LLDP results. 

### 3.2. Submicron Filtration Efficiency

The submicron filtration efficiency, a crucial parameter in the design and performance evaluation of filtration systems, has been determined according to the French standard AFNOR NFX45-104 [23] using as the test fluid ultrapure water solutions of traceable latex microspheres having a narrow size distribution of mean diameters ranging from 0.1 to 1 µm (0.05 µm particles were also used for the UF filtration device (F)). The experimental setup arranged for the submicron efficiency determinations is shown in Figure 2.

The test consists of dead-end filtering through the tested membrane filtration devices, a suspension containing certified monodimensional latex particles. Measuring the concentration of the suspensions of calibrated particles of the feed and permeate (particle counting systems are placed upstream and downstream the tested filtration devices, see (7) in previous figure) allows us to calculate, for each class of particle sizes of latex, the particulate filtration efficiency. The counting was performed online by laser diffraction counting the monodispersed latex spheres suspended in the ultrapure water (obtained through a Milli-Q^®^ SimPak2, Merck KGaA, Darmstadt, Germany, cartridge) in accordance with the ISO 21501-3(2019) standard [24].

The submicronic filtration experiments were conducted using the following experimental conditions: Test fluid: Ultrapure water, Milli-Q^®^ SimPak2 cartridge (Merck-Millipore).Flow rate: 2 L/min for MF/UF filters.Temperature: 23 °C (±2 °C).Particles: NIST^®^ traceable latex beads, from Thermo Sci^®^, with a mean diameter between 0.05 and 1.0 µm.Test end criteria: On-line particle counting where larger sizes of latex beads are not tested if 3 successive sizes give 100% filtration efficiency.Injected volume flow: From 2 to 30 mL/h.Counting circuit: Mod HSLIS-M50 at 100 mL/min (flow rate) during 60 s and size from 0.05 to 0.2 µm; Mod HSLIS-M100 at 300 mL/min (flow rate) during 60 s and size from >0.2 µm to 1 µm.

### 3.3. Liquid–Liquid Displacement Porometry (LLDP)

The fluid–fluid porometer (FFP; model PRM-8710^®^) developed by the IFTS consists of an automated pressure constant device suitable for gas/liquid and liquid/liquid tests (Figure 3). It has been developed and commercialized by the IFTS, and it is right now the only complete fluid porometer (GLDP and LLDP) device available on the market [18]. The device is configured in the LLDP mode for testing pore sizes down to 4 nm and requires relatively low pressures for the characterization of porous membranes in the tight UF range, covering also normal UF and MF. The equipment allows for implementing very stable pressure and leads to very accurate measurement of the resulting fluxes by using an analytical balance (Sartorius^®^ 6202i, accuracy ± 10 mg and range up to 6 kg). The porometer, which includes easy control and operation and automatic data acquisition, is provided with software which can determine (from experimental data) several important parameters related to the PSD, including the mean pore diameter, peak pore size, pore size distribution, solvent permeability, and droplet point. It also can be adapted to analyze various membrane types, including hollow fiber, tubular, and flat sheet, and several types of filtration devices. With these modifications of the original IFTS porometer, it is now able to test by means of LLDP small-sized filtration devices with no need to destroy and detach the filtration devices to separate the inner membranes. 

The LLDP experiments were performed according to the usual procedure [25,26,27,28]. A very stable 1:1 (*v*/*v*) binary mixture composed of water/isobutanol (γ = 1.7 mN/m) was used in the experiments. The isobutanol used in the mixture was of reagent grade, and it was used as received without further purification (Sigma-Aldrich^®^: Darmstadt, Germany, Purity ≥ 98.5%). The temperature was kept constant at 22 °C (±0.1 °C). The mixtures were prepared by adding the targeted amounts of ultrapure water, Milli-Q^®^ SimPak2 cartridge (Merck-Millipore, Darmstadt, Germany) and alcohol into a separatory funnel and then shaking it vigorously before the mixtures were allowed to separate phases overnight. The water-rich phase (i.e., higher density phase) was firstly drained off, and then the organic-rich phase was collected, with both phases now fully immiscible with each other. The organic phase was used as the wetting liquid and the water phase as the displacing liquid because the alcohol-rich phase wets the polymeric structures better. Therefore, all the filtration devices were wetted in the isobutanol-rich phase for 30 min under a vacuum (approx. 150 mmHg).

The previously wetted filtration devices were then subjected to an increasing pressure of the displacing liquid at a predetermined rate. When the first droplets of pushing liquid appeared downstream, those droplets corresponded to the passage of the pushing liquid through the filtration device’s biggest pores. Further increments in the balance reading indicated an increase in the pushing liquid’s passage, and they were time collected to evaluate the flow of this liquid versus the applied pressure on each experimental step.

For each membrane filtration device, three porosimetric runs were performed on new freshly wetted filtration devices and the corresponding results were averaged.

### 3.4. Water Hydraulic Permeability

To check if the LLDP analysis affected the performance of the filtration devices, a simple measurement of the water permeability for all these filtration devices was performed after the LLDP run.

The water permeability measurements were carried out using the same IFTS fluid–fluid porometer (Figure 3), with water, provided by the Milli-Q^®^ SimPak2 cartridge, as the flowing fluid. Permeability measurement consists of imposing a constant pressure (from 0.1 to 3 bars) upstream of the membrane filtration device’s surface (dead-end configuration for flat and pleated filtration devices and out–in configuration for filtration devices containing hollow fibers) and increasing progressively the inlet pressure while recording the permeate water flow. The recording time was fixed at 30 s for each inlet pressure constant step. 

The hydraulic water permeability (L) was calculated by fitting data points to a linear relationship among the permeate flow (Q, in mg/s) and differential pressure (ΔP, in mbar). For all the filtration devices, regression coefficients over 0.99 were obtained.

### 3.5. Gas–Liquid Displacement Porometry (GLDP)

Previous characterizations (submicronic efficiency, LLDP and water permeability) were performed directly on fresh filtration devices as they are marketed so as to have the minimum influence on their actual performance. Nevertheless, to compare the results obtained, especially those of the PSD obtained via LLDP, with the other usually employed characterization techniques, some of the filtration devices (A, B and C, representing the three types of membrane configurations studied) were dissembled and the membranes inside were extracted. Small coupons of these membranes were cut to the appropriate size to be tested via the GLDP technique. It must be mentioned that the original filtration device cannot be directly tested via GLDP due to the very big gas fluxes obtained when the wetted filtration devices are emptied.

The GLDP measurements were also carried out using the same IFTS capillary flow porometer (CFP) (model IFTS-PRM-8710^®^) that is designed for working in both porosimetric modes (GLDP and LLDP). In the GLDP mode, the equipment is able to test pore sizes ranging from 0.3 µm to 500 µm. It allows for implementing very stable pressure (accuracy ±0.1 mbar), and the air fluxes are measured by means a mass flowmeter (accuracy ±1 mL/min). 

The sample coupons were first saturated in FC-43 (fluorocarbon 99% purity, supplied by 3 M and having a surface tension value of 16 mN/m). The displacing fluid consisted of compressed clean air steeply pressurized, starting from a very low pressure, and the gas flow across the sample was monitored. As the pressure was progressively increased beyond the bubble point, successive pores of decreasing sizes became gradually emptied and then contributed to the overall gas flow J_k_ through the sample, which was recorded for each applied pressure. The experience was followed until all the pores became empty of the wetting fluid and the successive flows became proportional to the pressure. The differential air permeability contribution at each pressure step was plotted in terms of the size of the pores yet opened, obtaining the corresponding PSD [28].

### 3.6. SEM Imaging 

The top surface of the small membrane fragments from filtration devices A, B, and C (detached from their respective filtration device housings) was examined using a Hitachi Tabletop SEM Microscope TM3000. Prior to imaging, the samples were coated with a graphite solution to enhance the image quality. Imaging was conducted at an acceleration potential of 15 kV, using magnifications ranging from 18,000× to 20,000× for each sample.

To analyze the SEM images, ImageJ software, a program developed under a permissive license at the National Institutes of Health (NIH), USA, was utilized. The black and white SEM images were digitized, and the pore sizes were measured by setting an appropriate grey level threshold to define what qualifies as a pore. Multiple images were measured for each membrane, and the resulting data outputs were averaged, with a minimum of 200 pores being considered for the analysis. This comprehensive approach ensured a robust and representative measurement of the pore sizes present in the membranes.

## 4. Results and Discussion

### 4.1. Submicronic Efficiency

All six water filtration devices underwent testing to evaluate their submicronic efficiency. In these tests, latex particles with mean sizes ranging from 0.05 to 1.0 μm were sequentially filtered through the filtration devices, and the retention of each latex particle size was determined. Figure 4 illustrates an example of the obtained results for filtration devices E and F. For the filtration devices falling within the microfiltration (MF) range (filtration device labelled A to E in Table 1), the particle retention was found to be nearly 100% (greater than 99%) for particles with a size of 0.2 μm and larger. On the other hand, the last filtration device (F), which was categorized as an ultrafiltration (UF) filtration device, exhibited a retention of 98.5% even for the smallest tested latex particles, with a size of 0.05 μm, and achieved 100% retention for larger latex particles. These findings highlight the high efficiency of the MF range filtration devices in retaining submicronic particles, while the UF filtration device demonstrated even greater retention capabilities for smaller particles.

The results of these determinations are shown in Table 2 for all the filtration devices. It reports the latex particle mean size which is fully retained (100%) in each filtration device. As expected, the UF filtration device leads to the retention of smaller particles, although all the MF filtration devices fulfil adequately the usual requirement of particle size retention for a filter designed for sterilization. It is considered that a filter exhibition retention cut of 0.2 μm can retain most bacteria present in water.

### 4.2. Water Permeability

In the last column of Table 2, the resulting values from the water permeability measurements conducted on all the cartridges after the LLDP analysis are provided. The resulting permeabilities exhibit a reasonable trend in relation to the mean pore size, with slight discrepancies attributed to variations in the cartridge size (not all the cartridges present the same dimensions).

Among the MF-UF cartridges, the one presenting the highest permeability (cartridge B) is also that for which LLDP measures the biggest mean pore size. Similarly, cartridge E presents both the lowest water permeability and the smallest mean pore size according to LLDP. 

Certainly, the lowest absolute values of permeability correspond to the UF cartridge (F), as expected. In fact, we can perform a simple calculation between the mean pore size and permeability of cartridges E and F. The ratio between the permeabilities of cartridges E and F is roughly 30, while the ratio between the corresponding mean pore sizes is 5.4 (whose square is almost 30), demonstrating the dependence of the permeability on the square of the mean pore radius, as predicted by the Hagen–Poiseuille model. In any case, the values of water permeability demonstrate that none of the cartridges have suffered any structural damage during the LLDP analysis that should lead to a considerably higher value of permeability.

### 4.3. Pore Size Distributions (PSDs) Obtained by LLDP

Figure 5 and Figure 6 illustrate two examples of the porosimetric LLDP runs conducted on a microfiltration (MF) filtration device (referred to as A) and an ultrafiltration (UF) filtration device (referred to as F), respectively. In both figures, the insert represents the LLDP run data, while the main plot (dotted lines) displays the resulting pore size distribution (PSD) in terms of the contribution of each pore to the membrane permeability, as calculated using Equation (2). Additionally, the PSD obtained after applying the Hagen–Poiseuille model for the convective transport of liquid inside the pores (Equation (3)) is represented as a vertical bar in the plot.

The first PSD, known as the permeability distribution, provides information about the contribution of each pore size to the overall flux through the filtration devices. On the other hand, the pore number distribution indicates the percentage of pores in each size class. As anticipated, the permeability PSD is slightly shifted to the right in comparison to the pore number distribution, reflecting the influence of the pore size on the overall flux characteristics.

These figures and the associated distributions provide valuable insights into the pore size distribution and permeability characteristics of the MF and UF filtration devices, enabling a better understanding of their filtration performance.

The PSD curves obtained for all the analyzed filtration devices, including those not shown here, facilitated the calculation of two mean pore sizes: one for the permeability contributions (d_p,avg_(fl)) and another for the pore number (d_p,avg_(nr)). The averaged values of these mean pore sizes, along with their standard errors, are presented in Table 2.

Importantly, the analysis of all the filtration devices demonstrated good reproducibility in the LLDP results, with minimal dispersion. This indicates that the proposed non-destructive test is accurate and maintains precision even when working with larger volumes compared to the typical analysis conducted on small membrane coupons.

### 4.4. Molecular Weight Cut-off Estimation 

Traditionally, the MWCO is determined from costly experiments concerning solute retention. However, these methods have evident drawbacks, including being time-consuming, resource-intensive, and subject to the influence of the experimental conditions. To overcome these challenges, a simple assumption can provide a reasonable estimation of the MWCO, once an accurate determination of the PSD has been obtained.

As explained in a previous work [29], we can use the pore size distribution (PSD) obtained via LLDP to evaluate the membrane’s cut-off value. This has been done for filtration device F, the only one in the UF range, leading to a value of 245 ± 20 kDa, which can be considered reasonable for an indirect calculation. This value is quite close to the upper range of UF, and it can be considered reasonable for a filtration device designed for the retention of bacteria and other organic molecules (as pyrogens or nucleases, as claimed by the manufacturer). Nevertheless, it results in a much higher value than the nominal value specified (20 kDa).

On the other side, the empirical correlation used in this estimation is based on the dimensions of a dextran molecule, which is frequently used in MWCO determinations. If we estimate the MWCO value from an empirical correlation valid for different molecules, for example, polyethylene glycol (PEG) [30], the resulting value (52 ± 5 kDa) is closer to the nominal value but still overestimates it.

We can go further by considering the value obtained for the limit radius (the one corresponding to 90% of the biggest pores) as the hydrodynamic radius of the molecule retained at 90%, while the actual molecule size should be the gyration radius. Considering the usual relation between both radii [30], the new estimation of the MWCO goes down to 24 ± 3 kDa, which now really is closer to the nominal value.

In any case, this example serves to highlight the difficulties of comparing the nominal MWCO values, as measured in experimental conditions and with test molecules normally not explained by manufacturers.

### 4.5. Pore Size Distributions (PSDs) Obtained by GLDP

Examples of the PSD curves obtained via GLDP are shown in Figure 7 for filtration devices A to C. The mean pore sizes obtained from the PSD for each filtration device analysis via GLDP are shown in Table 3.

The pore size distribution (PSD) analysis conducted via GLDP for the three membranes allows us to obtain the mean pore values presented in Table 3. These results will be compared with those obtained using other complementary porosimetric techniques in order to assess their consistency. Initially, it is worth noting that the results obtained via GLDP exhibit a reasonably good correlation with the nominal values, indicating the relative sizes of the pores in each sample. This correlation is expected, considering that GLDP is considered a reference method in the characterization of microfiltration (MF) membranes.

Moreover, the results obtained via GLDP show good reproducibility, as indicated by the relatively low standard deviation values (presented as error estimates) observed among the three measurements conducted for each sample, as shown in the tables.

Comparing the values obtained via GLDP and LLDP, we find a reasonable agreement, particularly in terms of the flow distribution values. However, there are slight discrepancies for membrane A, and both porometries slightly overestimate the manufacturer’s values for membranes A and B. It is important to consider that these filters are designed for water sterilization, resulting in nominal values consistently close to 0.20 μm, even though their actual pore size distributions may vary slightly.

### 4.6. Pore Size Distributions (PSDs) Obtained by Image Analysis of SEM Pictures

Figure 8 shows examples of the SEM pictures acquired for the membranes coming from filtration devices A, B and C. 

From these and similar pictures at the same magnifications, ImageJ software was applied to determine the PSD of the three analyzed membranes. Firstly, it must be pointed out that in this case, we are talking about a surface PSD, since microscopic images cannot study the interior of the pores. This fact would be relevant for this type of filter, where bacteria retention is usually obtained via a deep filtration structure, with smaller pores in the interior of the membrane structure than in the actual surface.

In any case, the PSD obtained for membranes A and C by image analysis are shown in Figure 9. The distribution is given in terms of the Feret diameter (which corresponds to the shortest distance between two parallel planes limiting the pore image). In both cases, the distribution of the pores is much extended, including very large pores. This is a common fact when working with image analysis, which very often can match as pores simple darker areas on the surface. But while the PSD for membrane A can be reasonably matched with the expected pore sizes (with a mean value of around 0.3 μm, a gentle population of much smaller pores, surely attributable to unevenly illuminated areas, and a large tail of bigger pores), this observation is not true in the case of membrane C, where most of the population, including the apparent mean pore size, are much higher than the expected values (1 μm and over).

Similarly, the PSD obtained for membrane B, while not presented here, is quite far from the expected values, showing pores much greater than those corresponding to a 0.22 μm nominal filter.

The reason for the notable discrepancy observed is attributed to the presence of a substructure within these membranes, as characterized by larger voids and smaller pores. This substructure can be clearly observed in Figure 10, which is an enlargement of the picture of sample C depicted in Figure 8. It is within these smaller pore sizes that the true selectivity of the membranes resides, as demonstrated in the aforementioned figure. To ascertain the apparent diameters of several of these pores, a manual determination was conducted without utilizing image analysis tools such as ImageJ. This operation was also carried out for the images of samples A and B, and the average values obtained from manually measuring the size of 10 pores in each image are presented in Table 3.

### 4.7. Comparison of the PSDs Obtained by Different Techniques

Finally, Figure 11 represents the resulting LLDP mean diameters (permeability and number) for all six filtration devices with their respective error ranges. For a clearer comparison, the respective nominal values (when available) are also marked as blue lines. As can be seen, the agreement between the nominal and actual mean pore size values is not complete, with a clear sub-estimation of the nominal values in the case of filtration device B and much closer agreement for the rest of the MF filtration devices. In the case of the ultrafiltration (UF) filtration device (F), it is common practice to specify the MWCO rather than the nominal pore size.

The comparison between the LLDP results and nominal data reveals a reasonably good agreement in several instances, particularly with regard to the mean pore size obtained via LLDP in terms of the flow (d_p,avg_(fl)). However, this comparison is less favorable for the number of pore-based PSD (d_p,avg_(nr)). It is expected to observe a higher discrepancy in this case, as the pore number distribution relies on assumptions about the transport mechanism through the pores, whereas d_p, avg_(fl) is directly derived from experimental data.

Furthermore, Figure 12 illustrates the mean pore size values obtained using various porosimetric techniques employed in this study. While this comparison is applicable only to membranes A, B and C, since they represent examples of each membrane configuration, the results presented can provide a reasonable assessment of the reliability of each technique. The nominal pore size values provided by the manufacturers are displayed as dotted lines.

Upon initial inspection, a notable agreement can be observed between the nominal values and those obtained from the three porometries for membranes A and C. However, in the case of membrane B, all the porosimetric results consistently overestimate the nominal value. In any case, a good agreement between the different techniques is found, which reinforces the reliability of the newly developed porometer.

## 5. Conclusions

The non-destructive LLDP-based method here proposed proved effective in evaluating MF and UF membrane filtration devices for water purification. This technique yields promising results without requiring the dismantling of the devices. The results show low dispersion values and significant agreement with GLDP and other characterization techniques (GLDP or SEM images). All the analyzed filters proved appropriated for bacteria retention (MF devices) and the effective retention of smaller particles and pathogens (for UF devices), making all the devices suitable for water purification.

The differences between LLDP and other techniques can be attributed to factors like the swelling of the wetting phase in polymeric membranes and challenges in modeling capillary pore geometries, especially in highly interconnected pores like hollow fibers. Direct information obtained via LLDP, such as flux distribution data, helps overcome these limitations.

The LLDP modification for the online characterization of filtration devices prior to water sterilization remains valuable despite reasonable discrepancies (comparable to other porosimetric techniques). Traditional GLDP-based quality control requires high air flow, which is difficult with conventional porometers due to the large membrane areas in the devices. LLDP operates with lower liquid fluxes, enabling accurate online control without reducing the filtration area.

The proposed method reliably detects defects or pinholes in the analyzed filtration devices, which, if present, could impact permeate sterility, as shown through diffusion testing. In conclusion, the modified LLDP setup proves suitable and precise for the non-destructive characterization of membrane filtration devices.

A possible limitation of the apparatus is the maximum measurable flux, restricting the analysis of larger commercial devices. The technique shows promising interest for quality control, especially for filters with hollow fibers, which are difficult to assess with other techniques. The LLDP-based method effectively meets the demand for the non-destructive characterization of hollow fibers, simplifying the process by eliminating the need for individual fiber connections. The agreement in the MWCO estimation for the UF cartridges is clearly reasonable.

Overall, the LLDP-based characterization technique, as adapted to medium-sized commercial devices, has been successfully validated on a variety of samples. It allows for the online measurement of the porosity of MF/UF filtration devices, providing accurate and reliable results supported by sub-micron efficiency assessments. These promising results suggest a potential application in the industrial quality control of filtration devices and routine monitoring of performance decay due to membrane fouling decay.

## Figures and Tables

**Figure 1 membranes-13-00660-f001:**
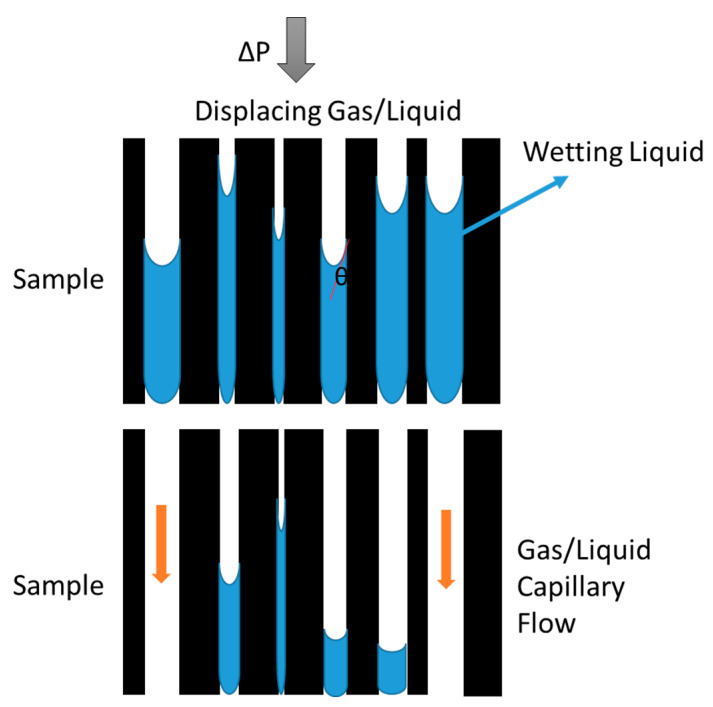
Capillary flow porometry principle.

**Figure 2 membranes-13-00660-f002:**
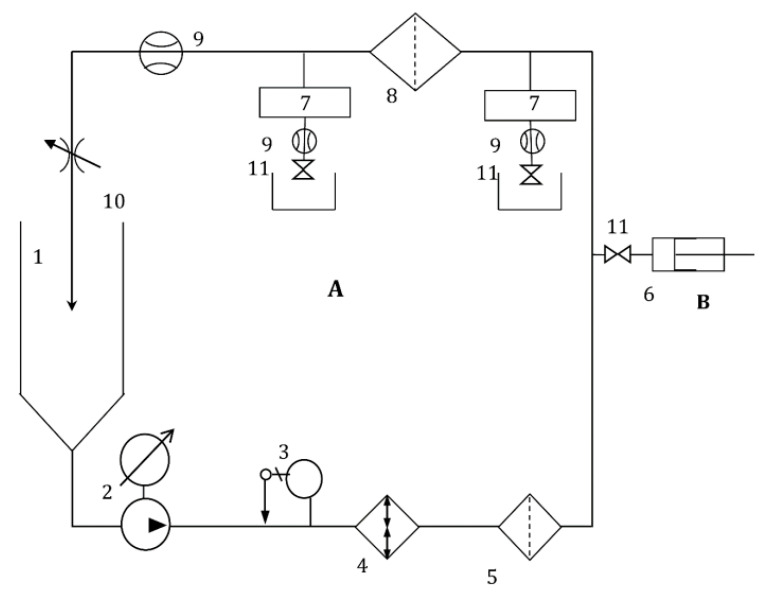
Schematic diagram of the submicronic test setup (A), with in-line counting of submicronic particles. The setup consists of the main reservoir (1), recirculation pump (LOWARA^®^, mod. CA70, from Xylem^®^, New York, NY, USA) (2), temperature regulator (3), heat exchanger (4), pollution filter (MF Pall^®^ Fluorodyne 0.45 µm and Pall^®^ Fluorodyne 0.02 µm, Pall, Saint-Germain-en-Laye, France) (5), particle counting systems (PMS^®^, mod HSLIS M100 and M50 (7), Boulder, Colorado, USA), test membrane filtration devices (8), flowmeter (Micromotion^®^, mod R-series, Rungis, France) (9), pressure regulating valve (KELLER^®^, Winterthur, Switzerland, type PR-33K) (10) and positioning valves (11). Latex particles were injected through the injection circuit (B), through an injection syringe (6), regulated by a valve (11).

**Figure 3 membranes-13-00660-f003:**
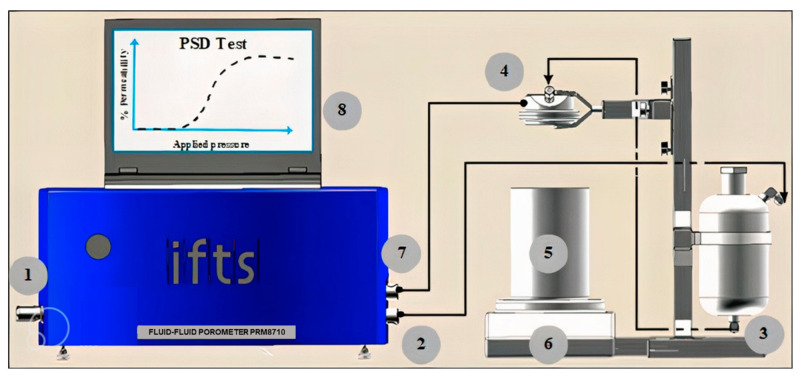
Schematic diagram of the IFTS porometer: (1) air supply; (2) air pressure line; (3) pushing liquid tank; (4) tested filtration device; (5) disposal; (6) analytical balance; (7) pressure sensor; and (8) computer, including the IFTS control and treatment software [23,24].

**Figure 4 membranes-13-00660-f004:**
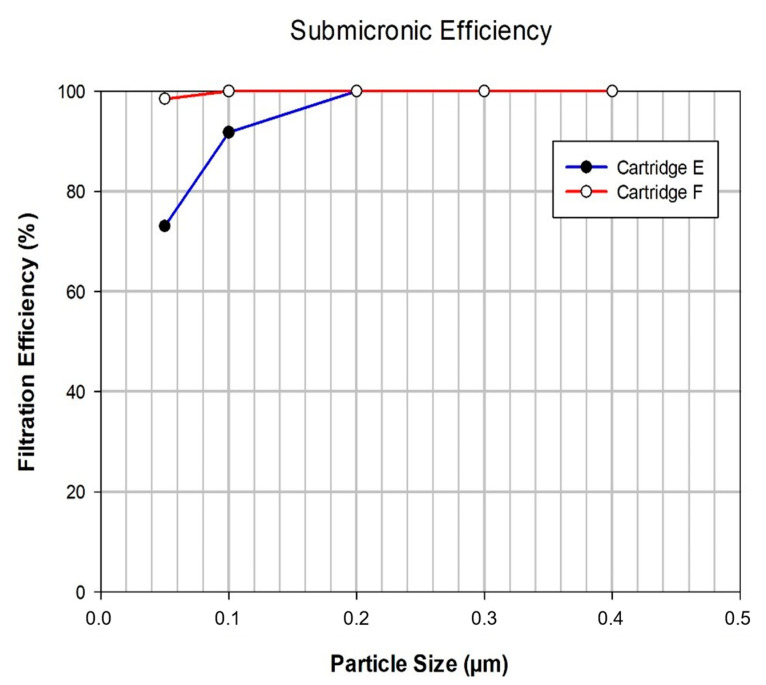
Example of the filtration efficiency curves obtained for filtration devices E and F.

**Figure 5 membranes-13-00660-f005:**
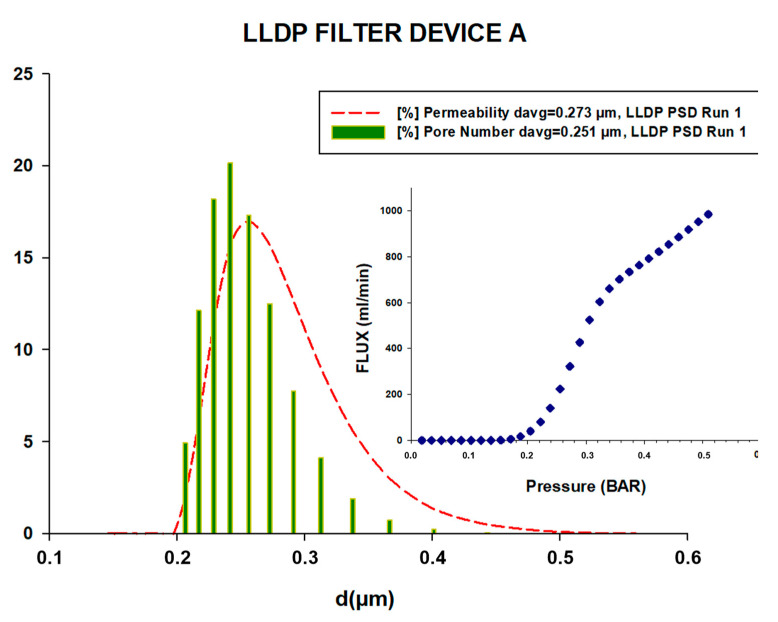
Example of an LLDP porosimetric curve and the resulting permeability and pore number PSD for filtration device A.

**Figure 6 membranes-13-00660-f006:**
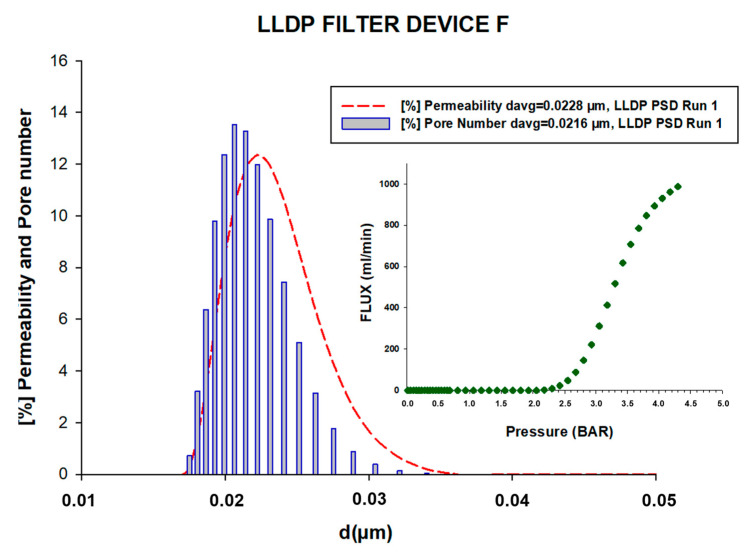
Example of an LLDP porosimetric curve and the resulting permeability and pore number PSD for filtration device F.

**Figure 7 membranes-13-00660-f007:**
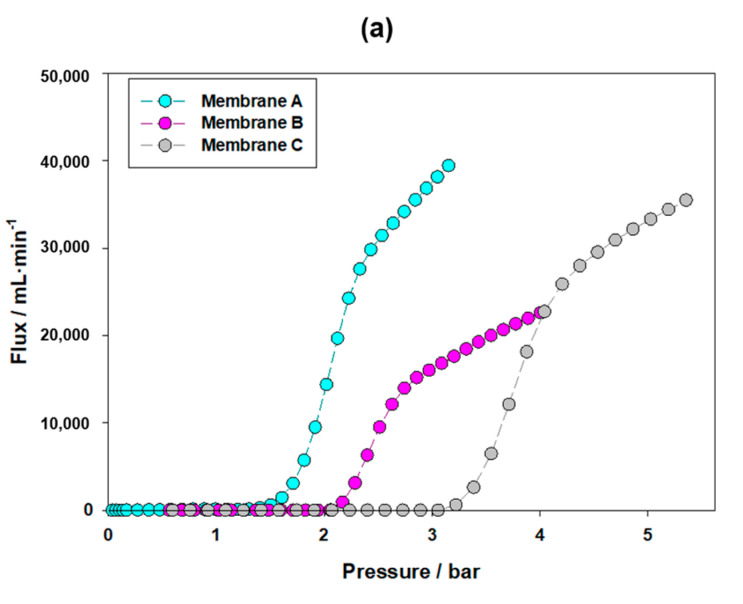
GLDP runs (**a**) and resulting PSDs (**b**) for an example of each filtration device (A to C).

**Figure 8 membranes-13-00660-f008:**
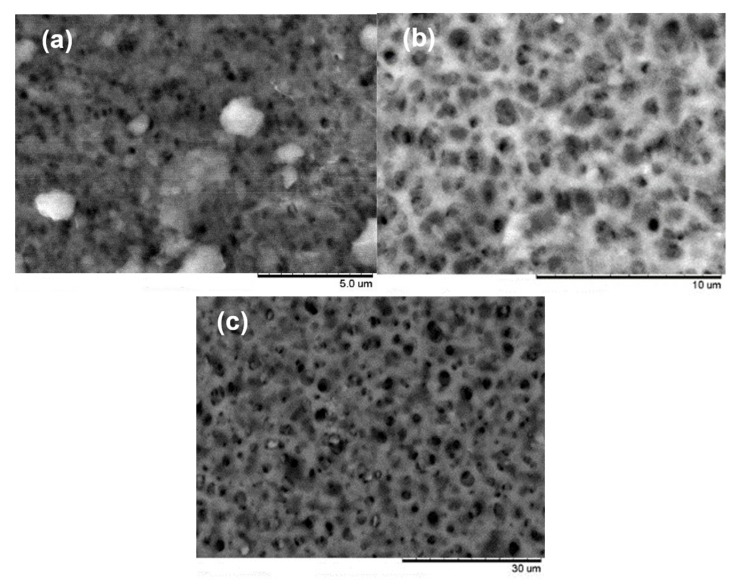
Examples of SEM pictures for the membranes in filtration device A (**a**), B (**b**) and C (**c**).

**Figure 9 membranes-13-00660-f009:**
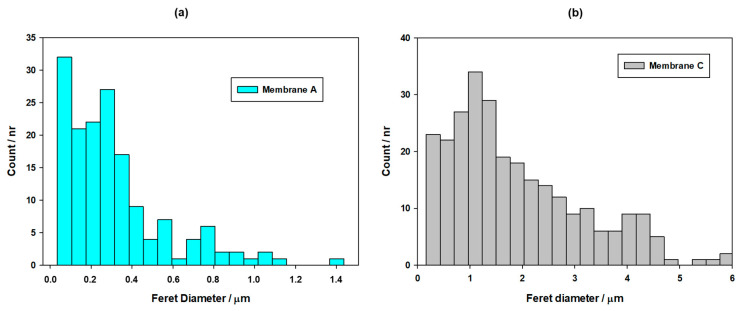
PSD obtained from the SEM pictures for the membranes in filtration devices A (**a**) and C (**b**).

**Figure 10 membranes-13-00660-f010:**
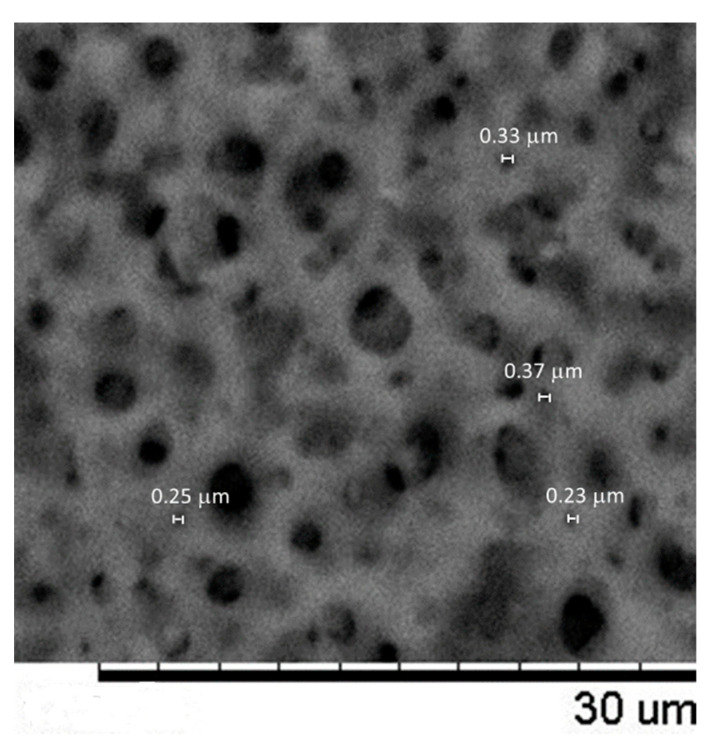
Amplified view of the SEM picture shown in Figure 9 for the membrane in filtration device C. Some pores presumably corresponding to the actual pore sizes have been individually measured.

**Figure 11 membranes-13-00660-f011:**
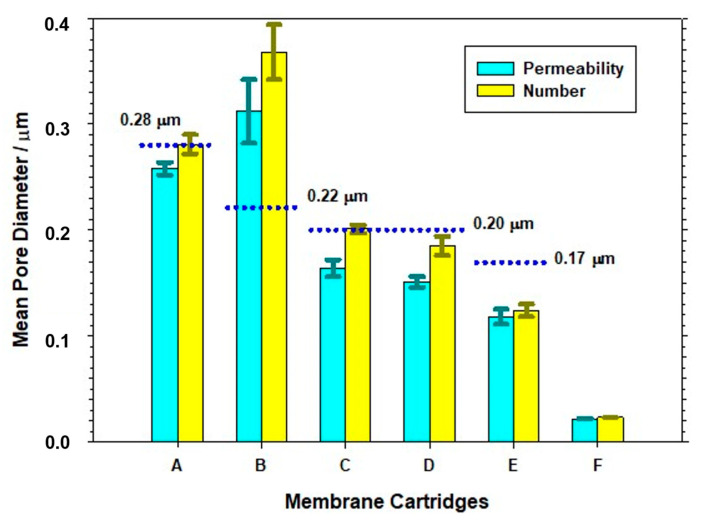
Comparison of the porosimetric results obtained via LLDP for all the studied filtration devices.

**Figure 12 membranes-13-00660-f012:**
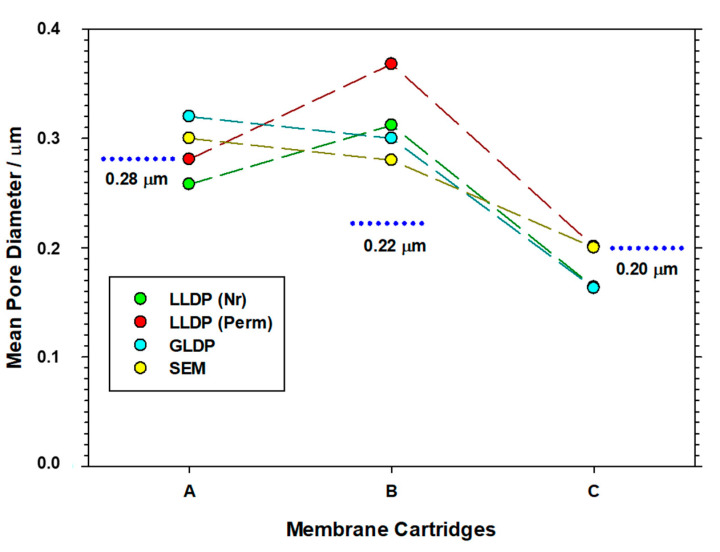
Comparative analysis of the porosimetric results for all the investigated filtration devices.

**Table 1 membranes-13-00660-t001:** Characteristics of the six membrane filtration devices analyzed.

Filtration Device Reference	Membrane Material	Nominal Pore Size/ MWCO	Pore Range	Type
A	PES	0.28 µm	MF	Hollow fiber
B	PES	0.22 µm	MF	Flat discs
C	Nylon	0.2 µm	MF	Pleated membrane
D	Nylon	0.2 µm	MF	Pleated membrane
E	PES	0.17 µm	MF	Hollow fiber
F	PS	20 kDa	UF	Hollow fiber

**Table 2 membranes-13-00660-t002:** Results of the characterization by submicronic retention, LLDP and water permeability of the six membrane filtration devices studied.

Filtration Device	Nominal Pore Size/µm	100% Submicronic Retention	LLDP d_p,avg_ (nr)/µm	LLDP d_p,avg_ (fl)/µm	Water Permeability (L/h/bar/m^2^)
A	0.28	>0.2 µm	0.258 ± 0.006	0.281 ± 0.009	1750 ± 30
B	0.22	>0.2 µm	0.31 ± 0.03	0.37 ± 0.03	2500 ± 100
C	0.2	>0.2 µm	0.164 ± 0.008	0.201 ± 0.004	2190 ± 60
D	0.2	>0.2 µm	0.151 ± 0.005	0.185 ± 0.009	1700 ± 80
E	0.17	>0.2 µm	0.118 ± 0.007	0.124 ± 0.006	1350 ± 60
F	-	>0.1 µm	0.0219 ± 0.0002	0.0231 ± 0.0001	45 ± 4

**Table 3 membranes-13-00660-t003:** Results of the mean pore size obtained via LLDP, GLDP and SEM for filtration devices A, B and C.

Filtration Device	Nominal Pore Size/µm	LLDP d_p,avg_ (nr)/µm	LLDP d_p,avg_ (fl)/µm	GLDP d_p,avg_ (fl)/µm	SEM d_p,avg_ (nr)/µm
A	0.28	0.258 ± 0.006	0.281 ± 0.009	0.32 ± 0.01	0.30 ± 0.07
B	0.22	0.31 ± 0.03	0.37 ± 0.03	0.30 ± 0.02	0.23 ± 0.05
C	0.2	0.164 ± 0.008	0.201 ± 0.004	0.163 ± 0.004	0.22 ± 0.06

## Data Availability

No other data than those published.

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
