# Peer review of "Automated Liquid–Liquid Displacement Porometry (LLDP) for the Non-Destructive Characterization of Ultrapure Water Purification Filtration Devices"

_membranes, 2023, doi:10.3390/membranes13070660_

Round 1

Reviewer 1 Report (Previous Reviewer 2)

I thank the authors to address my comments in the previous version. I believe the current version has been significantly improved. I would recommend acceptance of this manuscript. 

Minor language edits needed.

Author Response

Referee # 1

I thank the authors to address my comments in the previous version. I believe the current version has been significantly improved. I would recommend acceptance of this manuscript.

Thanks to the referee, your comments and those of the other referees guided us on improving the manuscript.

Reviewer 2 Report (New Reviewer)

The English language of the manuscript is good

Author Response

Referee # 2

Comments to the authors:

The authors used in all calculations Hagen-Poiseuille model for convective flow and it is not possible to make clear decision unless the authors answer the following comment:

  1. Line 180, equation 3, the authors stated that “the most suitable transport model is the Hagen-Poiseuille model for convective flow inside cylindrical capillary pores, so that we can assign a number of pores to each of these permeability increment values given by the following expression, [21].” Comments: is equation 3 used for straight cylindrical pore (ideal pore) or for real porous membrane (not cylindrical or straight pores). The Hagen-Poiseuille model for convective flow inside cylindrical capillary pores is not like equation
  2. There are two equations for Hagen-Poiseuille model for convective flow inside the pores as follows:
  3. For ideal straight cylindrical pores:
  4. For real porous membrane we should use the following equation:

There has been a misunderstanding in the equations quoted. We refer to the number of pores of each pore size class, those which should be needed to obtain the resulting permeability increase. In the first equation you quoted, N refers to flow of liquid, while for us, nk(d) is the number of pores having a given pore size. Considering permeability as the ratio between flux and transmembrane pressure (L=N/(P0-PL)), then your equation results in:

Which is very close to that we present in the paper (Eq. 3). The main difference is that eq. (3) refers no to the total number of pores in the membrane, but the number of pores in each class (Grabar and Nikitine derived an algorithm to convert discrete number of experimental points into a continuous function), which is given in terms of number · unit of pore radius. Converting the pore radius in diameter, appears the factor 2 which leads to the final appearance of our equation, totally consistent with that you propose.

On the other side, as you correctly explain, this equation is valid for straight cylindrical pores, an assumption quite common in converting experimental data into pore size distributions, as we have certainly commented din the manuscript (line 185). For more complex geometries, different modifications of general equation must be considered, among those that you comment, which is usually regarded as coming from the Carman-Kozeny model for flux in filters composed of an accumulation of grains of equal size (d). But any modification of ideal Hagen-Poiseuille model for cylindrical pores must be justified on a detailed knowledge of the actual pore structure in the inner of the filter (including the fact that very often this structure can change sensibly across the whole filter). That is the reason we consider the PSD based on pore number (that coming from application of Hagen-Poiseuille model) as less reliable, while the direct distribution based on flows (as directly application of only Young-Laplace equation) results more exact. Nevertheless, sometimes is important to complete the information of the flow distribution with that coming from number of pores, for example to get an estimation of MWCO, quite useful for UF membranes (section 4.4).

  1. Figures 1 to 9 are not clear, sharper figures should be added

Thanks for your suggestion, we have improved the quality of the figures. And we have included the original high quality pictures in a separate Power Point file to avoid any loss of quality in the pdf making process.

Reviewer 3 Report (Previous Reviewer 1)

I very much appreciate the authors efforts to revise the manuscript to address the concerns raised in my previous review.  I still have concerns about the novelty and significance of this work given the authors’ previous paper describing the automated porometer, but I will defer to the Editor regarding that issue. 

I had only a handful of minor comments:

1.     The writing is awkward in a number of places. For example, the Abstract currently reads: “Six commercially available filter filtration devices . . .”  There is no need for the word “filter” in this sentence. 

2.     Filtration device “A” is listed as “0.28 µm” for the nominal pore size. Is this correct?  I have never seen a filter listed with a pore size of 0.28 µm.

3.     In the procedures for the particle filtration experiments, the authors indicate that the injected volume is from 2 to 30 mL/h.  That is a flow rate, not a volume.

4.     Line 459 – This should be “devices A to C” instead of “devices A and C”.

5.     The SEM image of filtration device A in Figure 9 seems to show the presence of a number of particles on the filter surface. What are these?

6.     The x-axis of Figure 10 is labeled “Feret Diameter”.  The authors might want to define this.

Some minor editing would be helpful to improve the readability of the text.

Author Response

Referee # 3

I very much appreciate the authors efforts to revise the manuscript to address the concerns raised in my previous review.  I still have concerns about the novelty and significance of this work given the authors’ previous paper describing the automated porometer, but I will defer to the Editor regarding that issue.

Thanks for you kind comments. Regarding the novelty we have expressed (lines 113-119, Introduction and 657-662, Conclusions) that the main interest of the work is that the modification made on usual LLDP allows for characterizing small/middle size membrane cartridges without the necessity of breaking or destroying them, so the method can be used for a online control of cartridges quality.

I had only a handful of minor comments:

  1. The writing is awkward in a number of places. For example, the Abstract currently reads: “Six commercially available filter filtration devices . . .”  There is no need for the word “filter” in this sentence.

Eliminated filter. We have tried to improve the writing in some more sections (specially in conclusions section) to render it clearer.

  1. Filtration device “A” is listed as “0.28 µm” for the nominal pore size. Is this correct? I have never seen a filter listed with a pore size of 0.28 µm.

Strange but real , It is important to note that the data presented in our study were provided directly by the manufacturer. We understand that this may seem unusual, but we have diligently verified and validated the information to ensure its accuracy and reliability.

  1. In the procedures for the particle filtration experiments, the authors indicate that the injected volume is from 2 to 30 mL/h.  That is a flow rate, not a volume.

Thanks, corrected

  1. Line 459 – This should be “devices A to C” instead of “devices A and C”.

Thanks, corrected

  1. The SEM image of filtration device A in Figure 9 seems to show the presence of a number of particles on the filter surface. What are these?

Thanks for the comment. During the dismantling of the cartridge (using a destructive method) to extract the included membranes from the filter device (previously tested for LLDP or submicronic efficiency), the membrane pieces became eventually contaminated, as shown in the photo.

However, the authors made the decision to incorporate this photograph into the publication due to its relevance for comparison of PSD with proposed method. Obviously image analysis did not account for such dust spots.

  1. The x-axis of Figure 10 is labeled “Feret Diameter”. The authors might want to define this.

Thanks, a brief comment about Feret diameter has been added (line 539). Since the definition of pore diameter must be extended to irregular pores (not strictly circular) as measured by an image analysis software, the most common procedure is determining several Feret diameters (distances between two opposite borders of the pores, along a given direction) and promediate them (obviously this work is done by the software without any intervention from the user).

Reviewer 4 Report (Previous Reviewer 3)

This manuscript still has many issues. First of all, there is no reference cited in the entire section of Introduction. It is completely unacceptable. Besides, many sentences do not meet the scientific standard. For instance, Line 50-51. I have no ideal what the message the authors wanted to deliver. Furthermore, many figures are of poor resolution (Figure 6 and 7 particularly). I have raised this issue in the first round of review and yet no action was taken!

Table 1 – Why only 1 UF membrane was considered in this work? There are many different pore sizes of UF membranes and why limited it to only 1 UF. Authors on the other hand studied 5 types of MF membrane!

Figure 4 – Why need to show this photograph? It is not useful at all!

Abstract – Where are the key findings (quantitative data)? Authors completely ignored the key data in the abstract.

Conclusion – Conclusion should be short but concise. Obviously, the authors failed to meet the criteria. Rewrite all!

Extensive editing of English language required

Author Response

Referee # 4

This manuscript still has many issues. First of all, there is no reference cited in the entire section of Introduction. It is completely unacceptable. Besides, many sentences do not meet the scientific standard. For instance, Line 50-51. I have no ideal what the message the authors wanted to deliver. Furthermore, many figures are of poor resolution (Figure 6 and 7 particularly). I have raised this issue in the first round of review and yet no action was taken!

There has been a clear mistake in the introduction. In the manuscript there was 18 references corresponding to that section, as you can notice from the first appearing reference (line 124) which is quoted as 19. We have solved such mistake and now all original references appear properly included.

We have rewritten lines 50-51 to make clearer our point. The idea is that membrane processes have grown strongly over the decades of past century but water purification, as the initial application of synthetic membranes, is still one of the most important and demanded applications of membrane technology.

Figures 6 and 7, and the rest of them, has been improved in quality. Surely the problem comes from conversion to pdf, but hope now it has been properly solved. In any case we have sent to editors a Power Point including all figures in the maximum possible quality.

Table 1 – Why only 1 UF membrane was considered in this work? There are many different pore sizes of UF membranes and why limited it to only 1 UF. Authors on the other hand studied 5 types of MF membrane!

The main objective of the work was to demonstrate that the modification done in LLDP is useful for on-line analysis of water purification membrane filters. That is the reason we focussed on MF cartridges. The only UF cartridge included in the study, was considered to check if that proposed modification of LLDP still works for UF cartridges. Which is not strange as LLDP has been always considered more useful in the characterization of UF membranes.

Figure 4 – Why need to show this photograph? It is not useful at all!

Thank you for your feedback. After careful consideration, we have ultimately decided to remove the mentioned photograph from the publication. However, we would like to make a comment regarding the LLDP experiments. When working with two immiscible liquids, the original concept of bubbles, as observed in GLDP or the bubble point test, cannot be utilized to determine if the pushing liquid successfully displaced the wetting liquid and flowed through certain pores. The intention behind including the image was to demonstrate that the presence of droplets (not gas bubbles) on the outer surface of the membrane can serve as an indicator to verify the effectiveness of liquid displacement.

Abstract – Where are the key findings (quantitative data)? Authors completely ignored the key data in the abstract.

The aim of this work, as commented previously, was to demonstrate the ability of the technique to do a reliable, fast and non-destructive analysis of membrane cartridges. There was no real interest about the resulting pore sizes obtained but on comparing these sizes with other different techniques to check if our proposal is consistent. We think numerical results of the characterization are not really interesting the abstract, even when they are extensively discussed across the manuscript.

Conclusion – Conclusion should be short but concise. Obviously, the authors failed to meet the criteria. Rewrite all!

Thank you for your feedback. We sincerely regret the scarce clarity of the previous version of conclusions. Therefore, and taking your suggestion into consideration, we have revised the conclusions section and rewrite it to ensure it is succinct while effectively summarizing the key findings of our study.

We believe that the updated conclusions meets the desired criteria and provides a more concise summary of our aims, research and results.

Round 2

Reviewer 2 Report (New Reviewer)

Dear Editor

The authors answer my comments, therefore, I recommend to accept in Membranes

Regards

It is good

Reviewer 4 Report (Previous Reviewer 3)

I'm now satisfied with the action taken by the authors.

This manuscript is a resubmission of an earlier submission. The following is a list of the peer review reports and author responses from that submission.

Round 1

Reviewer 1 Report

This paper examines the use of a non-destructive liquid-liquid displacement porometry test for characterizing the pore size distribution of sterilizing grade filters.  Although the experiments appear to be carefully performed, it is unclear what is really novel (or helpful) in this work.  Liquid-liquid porometry has been used for decades to evaluate membrane pore size distributions.  The automated porometer used in this study has been described in a recent publication by these authors (reference 18). 

In addition to the above, it is unclear whether either the liquid-liquid porometry method, or the filtration efficiency data, are able to provide the type of characterization that is needed for sterilization filters (which are required to provide a sterile filtrate after channel with 107 colony forming units per cm2).  This would correspond to 99.99999% particle retention.  It is impossible to tell from the data in Figure 6 whether these membranes are able to provide that level of retention, nor is there any discussion in the paper as to the limit of quantification for either the filtration efficiency or liquid-liquid porometry experiments.  This information is critically important.

I also had a number of less significant comments:

1.     Based on the photos in Figure 2, several of the membrane cartridges appear to contain pleated membranes.  This is an important geometry for sterilizing grade membranes, but the authors never mention this in either the introduction, the experimental methods, or the discussion of the results.

2.     The caption to Figure 3 lists a series of membrane filters, including the Fluorodyne 0.45 µm and the Fluorodyne 0.02 µm.  However, neither of these filters is listed in the Materials and Methods, nor are they discussed anywhere else in the paper.  In fact, neither of these pore sizes is discussed anywhere else in the manuscript.  This needs to be corrected.

3.     Line 199 references Grabar for the expression used to evaluate the number distribution for the pores, but this reference is not provided in the Reference list at the end of the paper.

4.     As described in the text (lines 256), all of the filters were tested using the same volumetric flow rate (2 L/min), even though the filters have significant differences in membrane area.  Why weren’t the filters challenged at the same filtrate flux (volumetric flow rate normalized by the membrane area)?  This would seem to be much more appropriate, as well as more likely to give consistent behavior between the filters since it would lead to much better control of the extent of concentration polarization and fouling.

5.     The data in Table 2 present the water permeability in L/h/bar.  This is not meaningful unless one knows the area of the membranes.  The authors comment that the areas of the membranes are unknown, but all of the manufacturers must provide some information on the membrane area in their cartridges.  It would be much more useful if the permeability values could be normalized by the manufacturer’s provided membrane area, even if there are uncertainties in these values.

Typographical errors:

Line 40 – “0.2-0.22 mm” --> “0.2-0.22 µm”

Line 186 – “These increases can be dimensioned from” --> “These increases can be normalized by”

Table 1 – “Hollow fibber” --> “Hollow fiber” (throughout the table)

Line 224 – “fibbers” --> “fibers”

Reviewer 2 Report

This work by Peinador et al., proposed a new porosity method to charaterize microfiltration membranes. Admittally, this method is new but not novel, as it carries little meaningful insights to both industrial and academic community. However, from a perspective of publishing an effort to optimize or redesign a method to charaterize membrane, this manuscript comes with some values. I would recommend to accept this work as addressing minor comments listed below:

  1. Line 40, page 1, please be careful with unit, MF membrane is within micron not mm. Please recheck all other units used in the study.
  2. Please revise figures 1, 4, 5, 7, and 8, the quality of these figures are not acceptable, I could not even read the figures. Particularly for the figure 1, did you import this figure from other's work? If you did, please cite.
  3. Also, please use a well-accpeted method to also charaterize membranes, and compare with the results produced from your method. In this way, you could justify the validity and accuracy of the proposed method.

Reviewer 3 Report

The paper reported the use of LLDP technique in characterizing the properties of MF and UF membranes. However, the current stage of the manuscript does not meet scientific standard.

Authors should at least characterize these membranes using other method and compare the results obtained from LLDP to show the accuracy of the technique proposed. Authors should know that the pore size and other related parameters provided by the manufacturers might be not very precise and thus can be used to correlate in certain cases.

Introduction is very lengthy without key references to the main scope. Authors investigated LLDP, but its literature related to membrane characterization is very little.

Figure 1 is poor of resolution.

Table 1 – What is LRV? Besides, please provide the brand name for each cartridge and how does the manufacturer determine the pore size (the technique).

Figure 2 – Please provide the dimension of each cartridge.

Figure 5 – It is hard to understand the figure. Authors should present it professionally!

Figure 6, 7 and 8 – All the data of 6 cartridges must be provided (At least to include in supplementary file).

Table 2 - The water permeability unit is in L/h/bar. This unit is meaningless without having area (m2). Besides, why the value of F is 10 times lower than A? What are the reasons? Discuss it.

Figure 9 – The legend is about permeability and number, but the y-axis is mean pore diameter. Why is the figure about?